# Green Urban (RE) Generation: A Research and Practice Methodology to Better Implement Green Urban Infrastructure Solutions

**Jon Laurenz [1,2,\*], Jone Belausteguigoitia [1,2], Ander de la Fuente [1] and Daniel Roehr [3]**

[1] Architecture # 137, School of Architecture, The University of the Basque Country,
20018 San Sebastian, Spain; jonebelauste@lurstudio.com (J.B.); ander.delafuentea@ehu.eus (A.d.l.F.)

[2] Lurstudio Koop, Elkarte Txikia, 48005 Bilbao, Spain

[3] Greenskinslab, School of Architecture and Landscape Architecture, University of British Columbia, Vancouver, BC V6T 1Z4, Canada; droehr@sala.ubc.ca

**\*** Correspondence: jonlaurenz@lurstudio.com

**Abstract:** Green Urban Infrastructure Solutions (GUIS) are becoming more and more popular globally. Recent research reveals the environmental benefits derived from GUIS as well as their contribution to climate change adaptation. However, the urgent need for GUIS in order to meet the Paris Agreement, has not translated into an easy implementation thereof. This paper proposes a circular design methodology (CDM) where the combination of research and practice contributes to minimize both current skepticism and barriers when implementing GUIS. It includes a community engagement process to better understand their sensitivity and build consensus on GUIS. Additionally, GUIS are implemented, in a series of pilot projects and specific research is applied to comprehend the environmental benefits derived from these GUIS. The paper argues that GUIS represent a significant opportunity to respond to climate change risks as well as to achieve other urban benefits; however, in order to overcome existing barriers and skepticism, the proposed CDM reaches for more consensual urban solutions and drives uptake and implementation of GUIS, contributing to move from pilot project to common practice.

**Keywords:** green urban infrastructure solutions; circular design methodology; climate change adaptation solutions; sustainable urban drainage systems; nature based urban solutions; stormwater management; low impact development; urban greenery; environmental benefits





## 1. Introduction

Urban Design today is experiencing important growth in green urban infrastructure solutions (GUIS). These types of solutions are effective tools in combating the impact of climate change [1]. An increasing number of examples can be found all around the world, as shown in the following recent projects from Copenhagen (Europe), Melbourne (Australia), and Bogotá (Latin America), Figure 1.

Furthermore, current research on this subject demonstrates that GUIS are cost-efficient policy tools [2] to respond to climate change. There is usually a high return on GI investments and overall reviews of restoration projects typically show cost:benefit ratios in the range of 3 to 75 [3]. Research also demonstrates the environmental benefits derived from GUIS (references included in Appendix A).

However, despite this increased implementation and research of GUIS, there is still resistance to these solutions. A series of pilot projects, carried out by the authors, reveals the barriers encountered when implementing GUIS in practice. Some of the most prominent barriers are the lack of experience in GUIS and overall skepticism regarding innovative urban solutions.

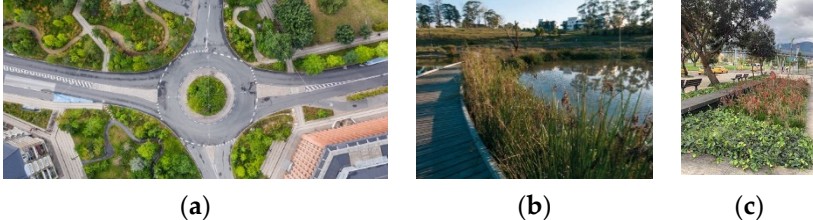

(**a**)            (**b**)            (**c**)

**Figure 1.** Examples in Copenhagen (**a**) https://www.sla.dk/en/projects/bryggervangen-sankt-kjelds-plads (accessed on 19 April 2021); Melbourne (**b**) https://www.melbournewater.com.au/sites/default/files/South-Eastern-councils-WSUD-guidelines.pdf (accessed on 3 May 2021); and Bogotá (**c**) https://www.cra.gov.co/documents/Cuaderno-Regulatorio-2vf.pdf (accessed on 23 November 2021).

In addition, the International Paris Agreement of December 2015 [4] was a turning point in the international commitment of countries around the world to strengthen the global response to the threat of climate change. Moreover, the IPCC Climate Change Report of October 2018 and of August 2021 [5,6], called for urgent action in order to limit climate change impact. Considering both the Paris Agreement and the IPCC Reports there is a pressing need to broadly implement GUIS as a response to climate change. GUIS, at the building scale, contribute to mitigate climate change in terms of reducing $CO_2$ emissions, due to their contribution to energy efficiency in buildings, Table A1. GUIS, at the urban scale, are effective climate change adaptation solutions, which mainly reduce urban heat island and stormwater runoff (and consequently, floods, Table A2). This means all barriers to implementation of GUIS should be necessarily overcome.

In this context, a circular design methodology (CDM) is proposed in order to minimize current barriers and drive greater acceptance when implementing GUIS. CDM demonstrates that it is an effective process to better implement GUIS in practice, in terms of gaining community acceptance, support, and consensus for the proposed GUIS. Consequently, existing skepticism and resistance to GUIS are minimized.

## 2. Circular Design Methodology (CDM)

CDM is based on three main phases: research, community engagement, and implementation (professional practice). It must be noted that each phase feeds into the next phase generating a circular cycle (Figure 2).

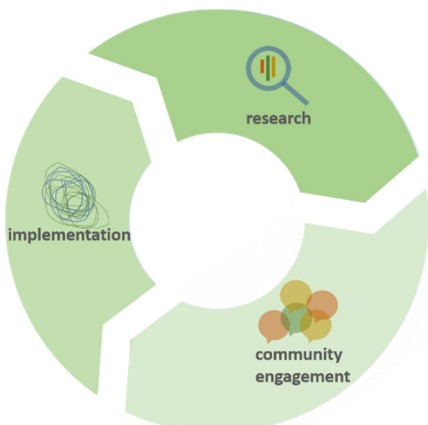

**Figure 2.** Scheme of the proposed CDM (image by the authors).

The CDM first (research phase) analyzes different existing GUIS in literature in order to identify, classify, and conceptually frame them. It classifies them into two main categories: green urban infrastructure at the building scale, and green urban infrastructure at the urban scale. Each of these two main categories is divided into a series of subcategories, described

in Appendix B. This research is applied to each specific project in order to identify the GUIS relevant to the site.

Subsequently (engagement phase), the methodology proposes a community engagement process as part of a collaborative design process to build social consensus and gather contributions regarding the urban proposal.

After the community engagement process, the GUIS are revised, accepted, and implemented into the design project (implementation phase).

Additionally (new research phase), further research explores the contribution of the selected GUIS to different environmental benefits in their site-specific context; i.e., benefits in terms of mitigation and adaptation to climate change, such as reducing the effect of urban heat islands and stormwater runoff, contributing to the energy efficiency of buildings, and reducing $CO_2$ emissions. An analysis of the contribution to other urban parameters, such as human comfort and psychology, air quality, urban aesthetics or urban biodiversity, and agriculture, is also included.

Finally (new phase of implementation), taking into account the results of previous CDM phases, the selected GUIS are implemented and built on site. The lessons learned from these pilot projects are explained in order to improve further GUIS implementation.

Findings stemming from the proposed methodology (CDM) showcase CDM as an effective tool to drive and uptake a consensual implementation of GUIS. In addition, CDM can be replicated in any urban project and scaled up to larger urban interventions. The authors argue the potential of CDM to widely implement GUIS in any urban development.

The different phases of CDM are explained in more detail in the following sections.

## 3. Research Phase

CDM begins with the research phase, which takes place in two different stages. First, it includes the analysis and classification of GUIS (more detailed in Appendix B) as well as the environmental benefits derived from GUIS (Appendix A). Secondly, a specific study of these environmental benefits is developed when implementing GUIS at a particular site.

The first research stage is applied to each specific project in order to identify the GUIS relevant to the site. Thus, the first initial analysis and classification of GUIS is used in the next phase of the CDM, that is, the community engagement process, in order to inform and train participants on the subject, as well as to identify the GUIS relevant to the site.

The second stage of this research phase occurs after the implementation of GUIS in the design project (professional practice phase). It applies the related research on the environmental benefits of GUIS to the specificities of the site providing detailed data on these environmental achievements.

## 4. Community Engagement Phase

The community engagement phase is the core of the proposed CDM. It is key to collaboratively design and support GUIS implementation. Many cities currently pursue mitigation and adaptation solutions to climate change. However, many citizens resist the implementation of such policies and other mitigation initiatives in their own neighborhoods [7]. Nevertheless, citizens have much to contribute to the successful design and implementations of GUIS. It is critical for this phase to include and build local community awareness and community acceptance regarding the collaboratively selected GUIS. From consultation to collaborative planning and active participation, citizens can become involved at all stages of a nature-based solution [8]. Specific research suggests that, if individuals are provided with user friendly, understandable, and relevant information and visualization about why and how green urban interventions can help respond to climate change, their understanding and acceptance of such interventions may improve [9,10]. In addition, by establishing which green infrastructure features are shared across groups, urban planning processes can successfully integrate public subjectivity into decision-making processes and respond to user preferences [11]. By doing so, urban design and planning will pro-

mote more active citizens, stakeholders, and governments, thus contributing to energy citizenship [12].

In this context, the authors developed a community engagement framework that seeks to better inform and raise understanding of GUIS among the community, and to build social acceptance towards them. This framework was applied in three participation processes undertaken in parallel in three Basque municipalities (Amurrio, Legazpi and Balmaseda) throughout the months of March, April, and May 2018, as part of the LIFE—Good Local Adapt European Project [13].

### 4.1. Objectives of the Community Engagement Process

The main objectives of the three public engagement processes in Amurrio, Legazpi and Balmaseda were the following:

1. To raise awareness regarding the consequences of climate change at a local level, its impact and need (and opportunity) to adapt through the use of design and planning solutions, applicable to small and medium sized municipalities.
2. To identify the needs and opportunities of both traditional and innovative adaptation solutions, in relation to urban design and planning, building refurbishment, and water management, among others.
3. To contrast and prioritize with the local community the specific solutions that are to be implemented at a local level, in case of implementation (by public and/or private initiative).

In order to meet these objectives, the engagement framework included three phases: preparation, participation, and evaluation, as shown in the following timeline diagram (Figure 3).

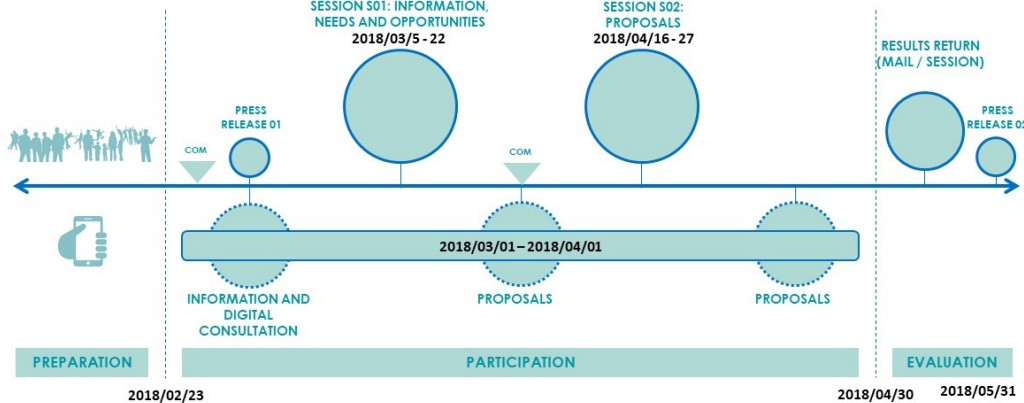

**Figure 3.** Engagement Framework diagram with the three phases: preparation, participation, and evaluation. Above the line, the main milestones for face-to-face participation Below the line, the main digital participation milestones (developed by the authors).

The engagement framework was the same for the three municipalities; although, each of them had its own particularities. For example, the communication channel to announce the participation process was different in each municipality. Legazpi announced it by placing posters on the doorways of the dwellings; Balmaseda resorted to its "Citizen Support Office" to inform its inhabitants; and Amurrio did so through its neighborhood associations. The following describes the framework synthesis of the comprehensive income and the resulting proposals.

### 4.2. Participation Process

The participation processes were developed in both face to face and digital formats.

### 4.2.1. Face-to-Face Participation

Face-to-face participation included two sessions (Figure 4), the first one (S01) being the analysis, with the objective of contrasting, agreeing on and prioritizing among participants the needs and opportunities for both building and urban design climate change adaptation solutions in the three demonstrative municipalities of Amurrio, Legazpi, and Balmaseda (Figure 5). The GUIS classification and general environmental benefits, analyzed in the research phase, were also included. In the second session (S02), the team presented the preliminary proposals (based on the results of session S01) and participants were asked to explore and develop them in further detail using proposal panels, an engagement tool designed by the engagement team, similar to the business canvas model, applied to engagement processes.

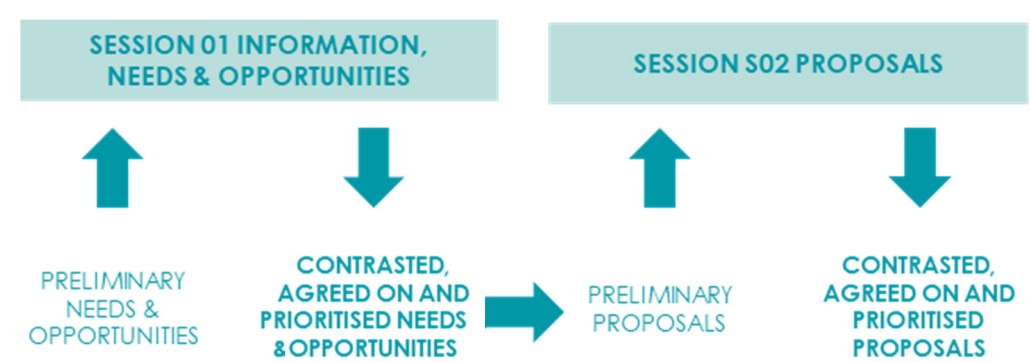

**Figure 4.** Face-to-face participation diagram (developed by the authors).

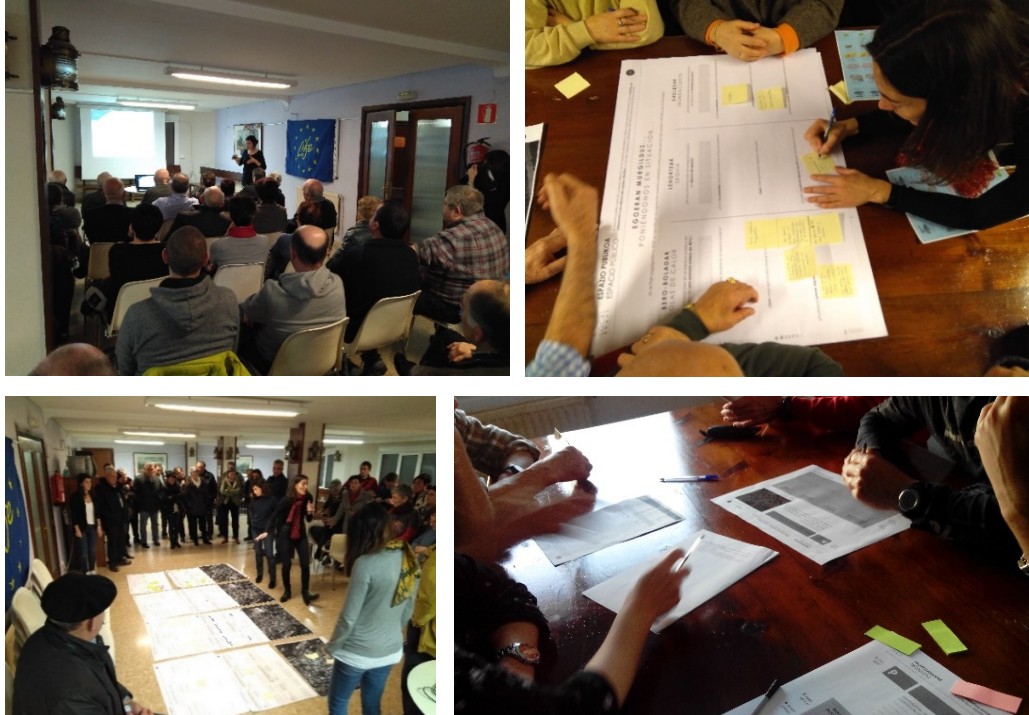

**Figure 5.** Images of the different sessions of the participation process (developed by the authors).

### 4.2.2. Digital Participation

Digital Participation was enabled through the on-line platform *Citizen* developed by one of the partners in the project team, Createlli, and accessible through the project website [13]. The participation platform was open to the public from 1 March to 15 May (2018), easily accessible by smartphone, tablet, and computer, with a comfortable, user-

friendly, and multi-language system (Basque and Spanish, the two official languages in the three municipalities) (Figure 6).

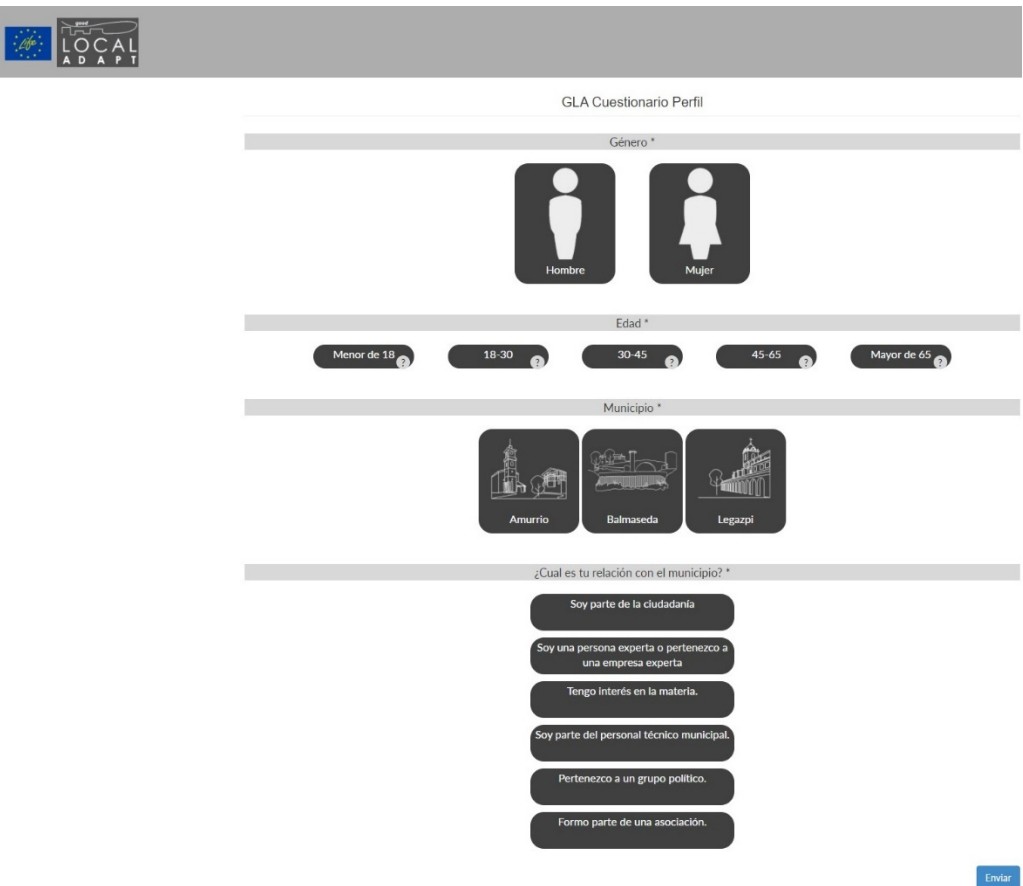

**Figure 6.** Screenshot of the on-line platform (Citizen): participants' profile.

Each of the three pilot case studies had its own digital participation site. After obtaining the participants' profile, the consultation was structured into each of the two main topics in the demonstrative neighborhoods: public space and buildings (Figure 7). The consultation method consisted of statements from participants in which they had to express a degree of agreement (from total agreement to total disagreement). Results were later expressed in consensus bars, which helped visualize the overall consensus (Figure 8).

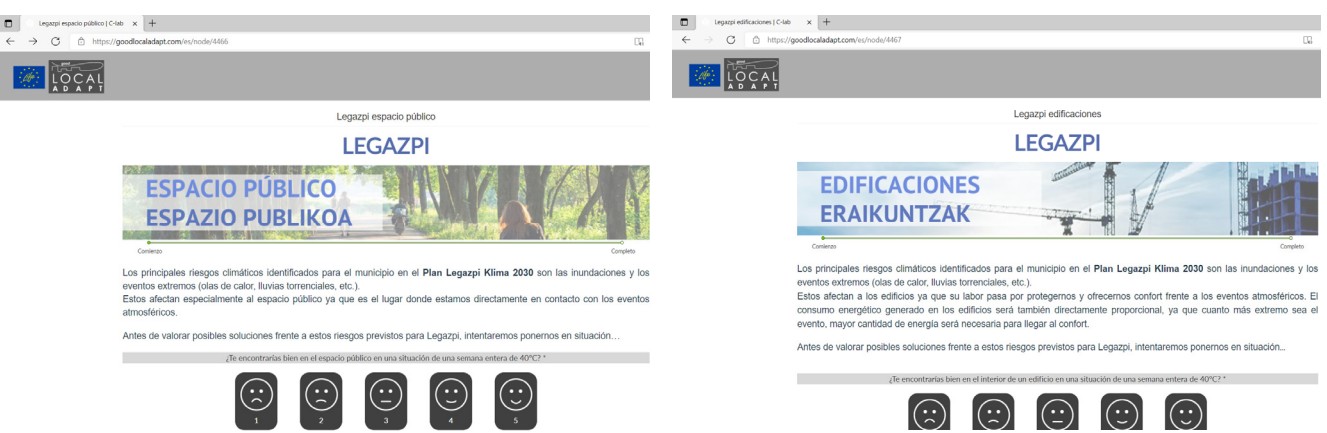

**Figure 7.** Screenshot of the consultation method of the on-line platform of Legazpi.

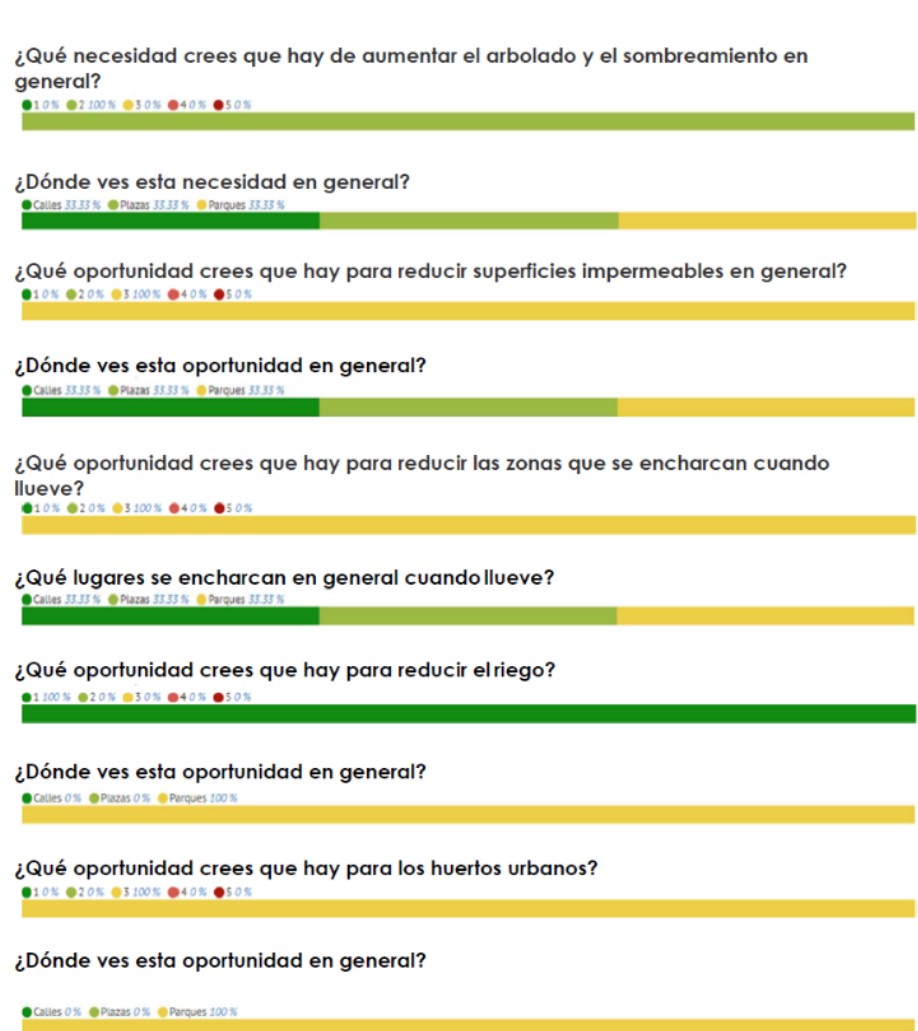

**Figure 8.** Consensus bars of the consultation method of Balmaseda.

### 4.2.3. Participation Indicators

The overall participation across all three municipalities amounted to 154 people, as shown in the table below. The main participation channel was face-to-face, with digital engagement being significantly lower. Legazpi was the municipality that registered the higher participation, probably due to the merging of this process with a local urban plan.

From a gender equality perspective, participation in all three municipalities was relatively balanced, with overall data showing 53% women and 47% men (Table 1). This balance is seen in all three municipalities, with small variations, the larger being in Amurrio, with a larger share of women participating. Of note is that these results are relevant in comparison with face-to-face participation in other engagement processes carried out at the time by the team in the Basque Country which revealed a less balanced makeup, with percentages of 33% women, 66% men.

**Table 1.** Participation by number, means, and gender (developed by the authors).

| Three Pilot Municipalities | Session S01 | Session S02 | Total Face-to-Face | Total Digital | Total |
|---|---|---|---|---|---|
| Number of participants | 79 | 67 | 146 | 8 | 154 |
| Women (%) | 51% | 57% | 53% | 50% | 53% |
| Men (%) | 49% | 43% | 47% | 50% | 47% |

The average age of participants was slightly different within the three pilot municipalities (based on visual observation in face-to-face sessions): in Balmaseda, ages ranged between 35 and 65; in Amurrio between 45 and 65; and in Legazpi, over 65. In digital participation, all three municipalities were aligned with an average participation age ranging between 30 and 45. In all three cases, participation of people under 30 was negligible or non-existent.

In the three pilot municipalities participants were mostly local residents, with slight particularities: In Amurrio, most people came from the overall municipality, with specific interest in the subject and, to a lesser extent, neighbors from the demonstrative neighborhoods of Landako and Goikolarra. In Balmaseda, participants were mostly local residents (both from the historic quarter and from the rest of the municipality) and, to a lesser extent, immigrants, technicians, and political representatives. In Legazpi, most participants were local residents from the four demonstrative neighborhoods: San Ignacio, San Martín, Arantzazu, and San José.

4.2.4. Results of the Participation Process

In session S01 participants prioritized the climatic risks for public space and buildings. In all three, the priority risk were floods, followed by draughts and heat waves (only one group, in Amurrio, differed, giving heat waves a higher priority than draughts). The needs and opportunities for climate change adaptation design solutions in public space according to participants in the three pilot municipalities are summarized below, in order of priority (Table 2).

**Table 2.** Needs and opportunities in public space (developed by the authors).

| Needs and Opportunities for Adaptation in Public Space 7 Demonstrative Neighbourhoods in the Three Pilot Municipalities: Amurrio, Balmaseda and Legazpi (from higher to lower priority) | Amurrio | Balmaseda | Legazpi |
|---|---|---|---|
| **1. Need to increase shadow (trees and vegetation).** | 3 | 1 | 1 |
| **2. Need to decrease impervious surfaces and create SUDS.** | 3 | 4 | 2 |
| **3. Opportunity to introduce urban gardens.** | 5 | 2 | 3 |
| **4. Opportunity to reduce irrigation.** | 6 | 3 | |
| 5. Need to redesign public space (Amurrio). | 1 | | |
| 6. Opportunity to generate air currents into public spaces (Amurrio). | 4 | | |
| 7. Need to improve and raise people's awareness on these topics (Amurrio). | 7 | | |

The needs and opportunities for climate change adaptation design solutions in buildings, according to participants in the three pilot municipalities, are summarized below, in order of priority (Table 3). The results correspond to Legazpi and Balmaseda, as Amurrio only focused on public spaces.

**Table 3.** Needs and opportunities in buildings (developed by the authors).

| Needs and Opportunities for Adaptation in Public Space 5 Demonstrative Neighbourhoods in the Three Pilot Municipalities: Balmaseda and Legazpi (from higher to lower priority) | Balmaseda | Legazpi |
|---|---|---|
| **1. Need to improve insulation in façades and roofs.** | 1 | 2 |
| **2. Need to improve insulation in windows.** | 3 | 1 |
| **3. Need to improve water management.** | 2 | 3 |
| **4. Opportunity to introduce vegetation in roofs.** | 4 | 4 |
| **5. Opportunity to introduce vegetation in façades.** | 5 | 5 |

### 4.2.5. Evaluation and Conclusions of the Participation Process

Respondents assessed the process very positively (Table 4). Out of the 35 questionnaires gathered in the three individual processes: 83% of the participants thought the sessions were useful for working on climate change design solutions that can be potentially applied in the municipality, and said they would recommend this type of processes for similar plans and projects; 97% of participants agreed facilitators communicated clearly and were easily understood; and 100% agreed participants communicated with respect among each other. Participants also made a few recommendations on how to improve the sessions.

**Table 4.** Evaluation of the participation process (developed by the authors).

|  | Agree | In between | Disagree |
|---|---|---|---|
| 1. This session has been useful. | 83 % (29) | 11% (4) | 6 % (2) |
| 2. Facilitators have communicated in a clear and easily understandable way. | 97 % (10) | 3 % (1) | 0 % |
| 3. Participants have talked with respect. | 100 % (10) | 0 % | 0 % |
| 4. I would recommend this type of processes. | 83 % (29) | 14 % (5) | 3 % (1) |

The three participation processes met the expectations, mainly due to the positive attitude of those who participated, and yielded specific proposals which were subsequently considered by the three towns and contributed to the GUIS projects.

At the same time, participants in session S01 also had the opportunity to see experiences from other parts of Europe and the potential replicability in their municipalities as well as the potential transferability of the solutions elsewhere in Europe. Participants confirmed that the project raised their awareness and understanding of climate change and they valued the process positively.

It must be noted that engaging local municipal technicians and political representatives as part of the group of stakeholders was critical for the subsequent application of the project outcomes in pilot projects and municipal ordinances.

## 5. Implementation Phase: Green Urban Infrastructure Solutions—Own Pilot Projects

The final phase of the CDM is the implementation phase which includes a series of pilot projects where the authors implement GUIS in each of the municipalities taking part in the LIFE-Good Local Adapt project (Amurrio, Legazpi, and Balmaseda) GUIS were identified and selected in the previous community engagement phase, and implemented in both at the building (Balmaseda) and urban scale (Legazpi and Amurrio). The lessons learned and conclusions reached by these implementations are also included.

### 5.1. Implementation at the Building Scale

GUIS implemented at the building renovation project in Balmaseda were a rooftop greenhouse; vegetated tiles; reuse of treated rainwater and grey water; and water sprinklers on the roof as a cooling system. (Figure 9).

In this case, GUIS were only implemented in the design project. Unfortunately, this renovation project was not executed due to external reasons (budget modifications caused by the pandemic). Nevertheless, the environmental and social benefits of GUIS are summarized in Tables A1–A5. In addition, the rooftop greenhouse allows for the incorporation of excess heating production to be re-directed to the building's own heating system, as well as reusing the excess $CO_2$ produced for plants in the greenhouse. However, the considerable extra weight as well as the costs may render this implementation prohibitive.

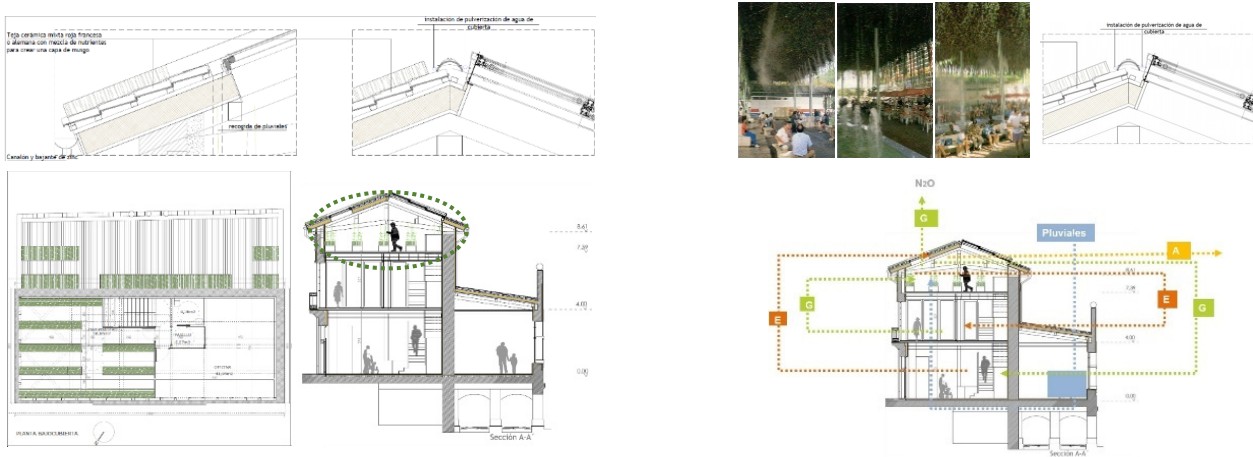

**Figure 9.** Vegetated tiles, rooftop greenhouse and water sprinklers in Balmaseda, Bizkaia (by the authors).

*5.2. Implementation at the Urban Scale*

In the two urban scale projects (Legazpi and Amurrio), the implemented GUIS were bio-retention areas; permeable pavements and shadings devices. Additionally, the project in Legazpi included a stormwater tank; and the project in Amurrio included green parking lot solutions (Figures 10 and 11). Unfortunately, the project in Amurrio was not built as a consequence of budget constraints.

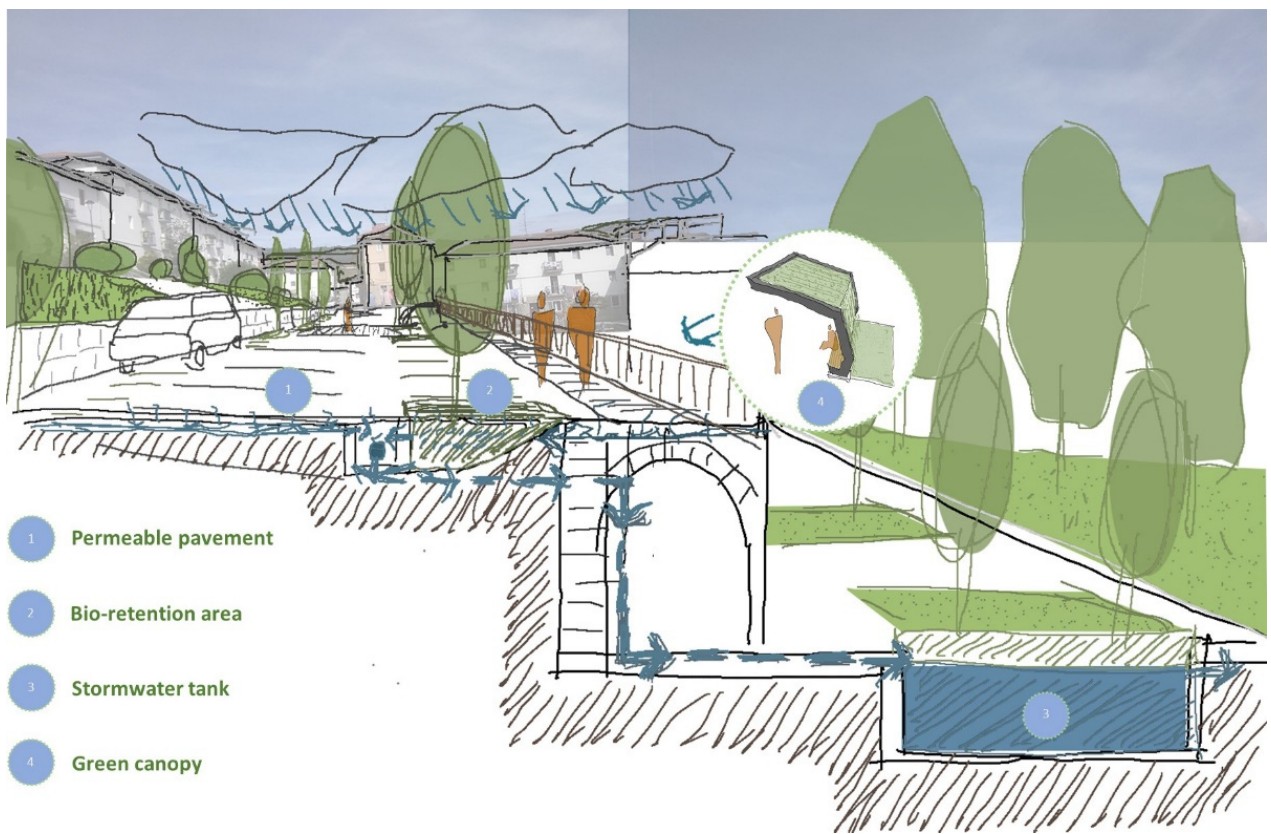

**Figure 10.** Scheme of the Legazpi's green urban infrastructure project (developed by the authors).

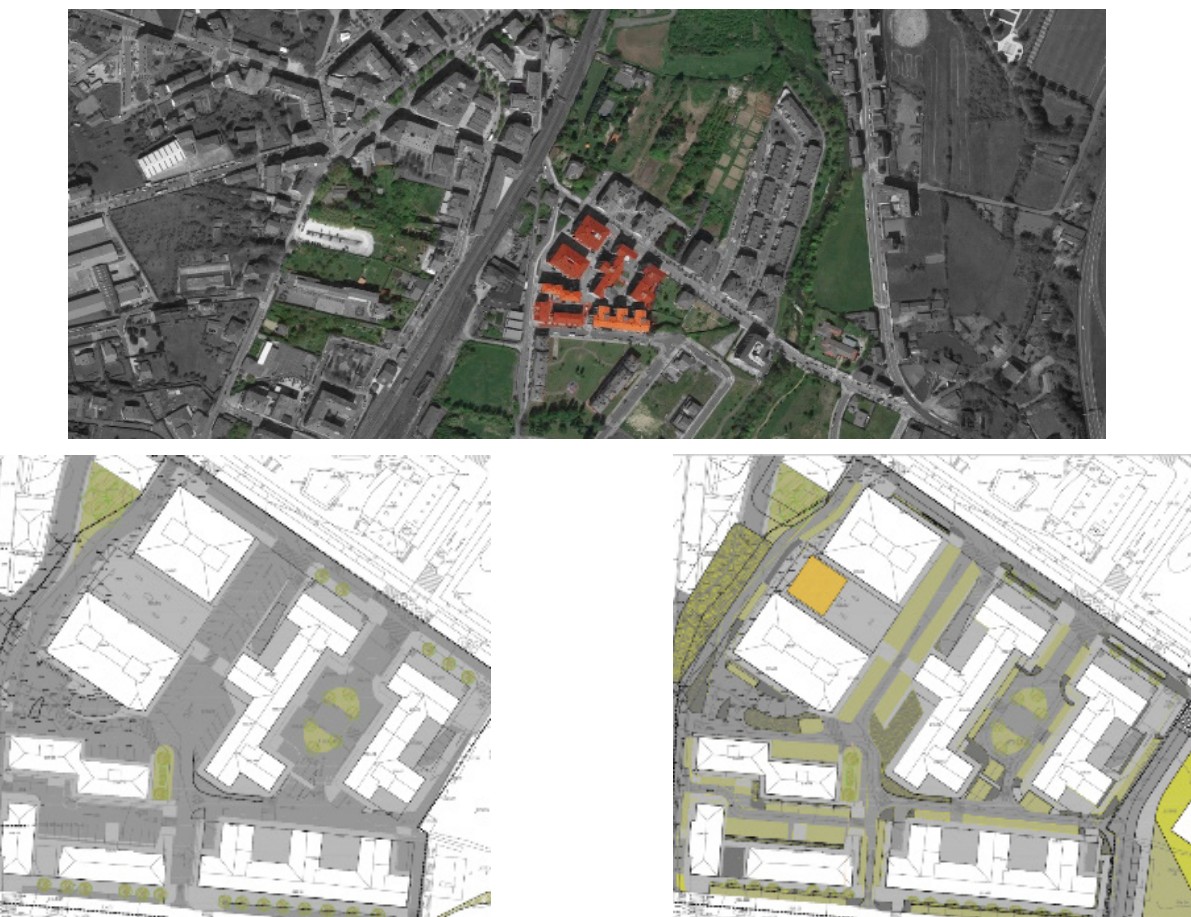

**Figure 11.** Drawings of the green urban infrastructure designs in (developed by the authors).

The project of Legazpi, as part of the second stage of the research phase proposed in the CDM, included an analysis of the specific environmental benefits derived from the GUIS implemented in Legazpi. This analysis revealed that the contribution of the proposed GUIS to the reduction in runoff from the site was approximately 26%; up to 20 °C to the reduction in the urban heat island; and 7% to the reduction in $CO_2$ emissions [14].

*5.3. Lessons Learned and Conclusions of the Implementation of GUIS in the Pilot Projects*

A relevant lesson learned from the implementation of GUIS in the Legazpi project was the importance of the impermeability study of the ground. This study changed the preliminary solution of directly infiltrating rainwater into the soil, to including a stormwater tank. Thus, the rainwater was guided to, collected, and retained in the tank to reuse it for irrigation or street cleaning.

Another important lesson learned was the construction system of permeable pavements. These require a special clean gravel base (5 cm maximum thickness; and 3–6 mm in diameter), instead of the conventional glue cement used in waterproof paving.

Regarding the selected stormwater tank. It was quite challenging to guarantee its impermeability. In addition, this paper considers this solution too artificial, and, for future practice, it advocates to retain rainwater through natural bio-retention areas rather than through artificial stormwater tanks.

Finally, the vegetated canopy should have included an irrigation system to ensure proper plant growth. A new moss wall and moss roof system were proposed. These are currently being tested and their suitability will be revealed over time. Nonetheless, an

irrigation system included from the beginning would have contributed to a better growth of the plant.

The conclusions from the Legazpi project show the importance of testing different GUIS in practice. There was a general lack of experience, both among the members of the design team and the administration technicians, as well as within the construction company. The financial barrier of GUIS should also be highlighted. In the authors' experience, only GUIS projects that receive financial support are implemented (the implementation projects described in this paper were funded by the European Commission through the LIFE GLA project [13]). However, results demonstrate remarkably greater environmental benefits than conventional solutions. Findings also highlight the suitability of the CDM methodology and its contribution to community acceptance for the proposed GUIS, and the enhanced implementation of GUIS.

The lessons learned from the two other design projects (Balmaseda and Amurrio, which were not executed), point towards the difficulties encountered when attempting to implement GUIS. The main barrier was the ongoing skepticism within the local government's teams, to embrace and promote GUIS. The fact that these are innovative systems, together with a lack of experience, result in decision makers unwillingness to take any risk.

Therefore, these experiences demonstrate that there is still a long, winding road ahead for policymakers to view GUIS as an attractive and reliable solution. This paper argues that CDM is an effective methodology to drive the uptake of GUIS by engaging both community and decision makers.

## 6. Conclusions and Discussion

A circular design methodology (CDM) is proposed in order to bring together research, community engagement and implementation, and give GUIS the drive they need. The use of CDM in three pilot projects reveals that it is a successful method to better implement GUIS in practice. The potential of this methodology relies on practice informed by research and research through practice, together with the key contribution to building social consensus and acceptance towards GUIS. In addition, it raises awareness and understanding of climate change adaptation solutions among the community, and also informs municipal technicians and decision makers about GUIS, their benefits, and contribution to improving their confidence in these types of systems.

The research phase of CDM shows the environmental benefits derived from GUIS. GUIS are effective solutions in combating the urban heat island; reducing stormwater runoff; with the results being more modest when it comes to reducing $CO_2$ emissions. In addition, the research phase includes other urban benefits derived from GUIS, such as human and urban comfort; psychological benefits; urban biodiversity; urban agriculture; air and water quality; landscape aesthetics, etc., which should also be considered.

However, the paper highlights the difficulties and barriers found when implementing GUIS in practice. There is still skepticism and uncertainty on how GUIS functions both among the community and decision makers. Therefore, in seeking to improve social consensus and acceptance towards GUIS, the community engagement phase of the CDM appears to be key. The participation process contributes to raising awareness and understanding of GUIS; it motivates community support for GUIS; and increases confidence among skeptics.

The implementation phase of CDM contributes to obtaining real experience of the GUIS execution. In addition, it provides valuable lessons learned from practice, as well as the barriers encountered when implementing them.

Thus, the research phase provides specific data to better understand the contribution of GUIS in reducing the impact of climate change. The engagement process achieves community awareness, consensus, and support to collaboratively proposed GUIS. The implementation phase helps enrich GUIS practice. Taken together, CDM minimizes existing resistance and skepticism, and facilitates opportunities for broader GUIS implementation elsewhere.

Currently, little or no research is conducted on these types of projects and, often, they do not include community engagement processes, nor specific information on GUIS and

their contribution to the impact of climate change. Therefore, current practice does not contribute to reducing the resistance and skepticism towards GUIS, which are the main barriers to GUIS implementation.

Thus, the authors propose CDM as an effective method to enrich the design of GUIS by raising awareness and understanding of GUIS among community and decision makers; reducing existing barriers by building social consensus and acceptance towards GUIS; and driving altogether a better implementation of GUIS.

CDM has been created to be transferable and applicable to different scales and places. In larger projects, more agents would need to take part in CDM, and the benefits from GUIS would be similar but would have a broader scope. However, further research is needed to assess and validate CDM's applicability to larger projects. Moreover, it must be borne in mind that using the CDM method may take longer than usual for GUIS implementation, but the benefits derived from its use, far justify this time investment.

Future research should also analyze whether CDM should include an urban regulation study to legally promote future GUIS. Valuable references to this type of urban regulation are the "Green Factors", which promote increased number of green areas in cities [15–17].

Urgent action against climate change consequences is crucial [4,5]. This 2020–2030 decade will be decisive to fulfill the commitments undertaken [18]. GUIS have the potential to respond to these challenges [1] and CDM is an effective method to accelerate a consensual implementation of GUIS, minimizing exiting barriers. It represents an opportunity to take action in order to move from pilot projects to common practice.

**Author Contributions:** Conceptualization and methodology, J.L.; community engagement analysis, J.B.; stormwater support and supervision, D.R.; supervising, A.d.l.F. and D.R. All authors have read and agreed to the published version of the manuscript.

**Funding:** The Project LIFE Good Local Adapt, included in the article as a case study, was funded by the European Commission through the LIFE 2016 program.

**Data Availability Statement:** Not applicable.

**Conflicts of Interest:** The authors declare no conflict of interest.

## Appendix A. Contribution of Green Urban Infrastructure Solutions to the Urban Environment (as Part of the Research Phase of CDM)

This section describes a series of research and compiles the findings extracted therefrom to better understand the contribution of GUIS within different urban parameters such as *mitigation and adaptation to climate change*; *human comfort*, *psychological benefits*; as well as the contribution *to air quality*; *biodiversity and urban agriculture*; *and architectural aesthetics*.

### *Appendix A.1. Contribution to Climate Change Mitigation*

Research demonstrates that green roofs reduce the energy demand of buildings on average by 10–25%, and consequently the $CO_2$ emissions [19–24]. Similarly green façades reduce the energy demand of the building between 10 and 30%; and the urban forest of a city reduces between 0.3 and 1% of the total $CO_2$ emissions emitted by that city [19,25–30] (Table A1).

**Table A1.** Summary table of the contribution to CC mitigation (developed by the authors).

| | Research | Topic | Contribution |
|---|---|---|---|
| **Green Roofs** | Ascione et al., University of Napoli and Sannio, Italy [19] | Green roofs in different European climates | - In warm climates green roofs reduce the air conditioning demand by 8–11%<br>- In cold climates, energy demand reduction by 4–7% |
| | Javier Neila, César Bedoya, et al. ETSAM UPM, Madrid [20] | Different types of green roofs in Madrid | - green roof with an integrated cistern reduces the energy consumption by 70% in comparison with an inverted roof<br>- thermal gaining reduction in summer by 60% in comparison with a non green roof |
| | Banting et al., Ryerson University for The City of Toronto [21] | Green Roofs in Toronto | - air temperature reduction by<br>- 0.5–2 °C<br>- energy savings: 4.15 $kWh/m^2$/year<br>- $CO_2$ mitigation: 32,200 tn/year |
| | Piero Bevilacqua et al., University of Calabria University, Italy [22] | Green Roofs in the Mediterranean | - not insulated green roof reduces demand by up to 34.7%<br>- insulated green roof reduces the energy demand by up to 25.4% |
| **Urban Forest** | McPherson, E.G. + Nowak, D.J. et al. [23] | Chicago's Urban Forest | - Reduces total $CO_2$ emissions of Chicago by 0.3–1% |
| | Valentin Scahefer et al. [24] | Urban Biodiversity | - shrubs trap 8.76 $kg/m^2$ of $CO_2$<br>- grasses, trap 4.38 $kg/m^2$ of $CO_2$<br>- vines/climbers trap 6.57 $kg/m^2$ of $CO_2$ |
| **Green Facades** | McPherson, E.G. + Nowak, D.J. et al. [23] | Chicago's urban forest | - Air conditioning reduction by 31%<br>- Heating demand reduction by 15% |
| | Schmidt, M. [25] | Transparent green facade in Berlín-Adlershof | - Total reduction of 100% in air conditioning with an external temperature of 30 °C, from that temperature air conditioning is necessary |
| | Laurenz, J.; Roehr D. [26] | Transparent green facade in Madrid and Vancouver | - In Madrid it reduces air conditioning by 45% and heating by 23%<br>- In Vancouver, it totally reduces air conditioning and total energy demand by 9% |
| | Mazzali, U. et al. [27] | Opaque green facade (like a tapestry) | - It reduces facades temperature by 12 °C in the north of Italy and by 20 °C in the center |
| | Perini, K. et al. [28] | Opaque green facade (like a tapestry) | - It reduces the air conditioning by 26% |
| | Perez, G.; et al. [29] | Opaque green facade (like a tapestry) | - It reduces facade temperature by 5–17 °C and increases the humidity by 7%<br>- It reduces the air conditioning demand by 20–30% being able to reach 50% |
| | Olivieri, F.; et al. [30] | Opaque green facade (by panels) and transparent green facade | - Opaque green facade reduces facade temperature by 5 °C and air conditioning by 12%<br>- Transparent green facade reduces facade temperature by 3–4 °C and increases humidity by 0% |

*Appendix A.2. Contribution to Climate Change Adaptation*

The analysis focuses on the contribution of GUIS in reducing urban heat island and rainwater runoff (and consequently reducing flood risks). Findings show that urban greenery maintains outdoor air temperature, and may even reduce it by 0.5–2 °C [31–35]. Consequently, urban greenery considerably reduces the temperature of paved surfaces to around 7–20 °C [36,37]. Regarding the reduction in runoff, green roofs retain 70–90% rainwater in summer and 40–50% in winter [21]. Green façades retain 50–75% and can reach 100% [37]. A global green urban infrastructure strategy can retain 90% of a city's runoff [38] (Table A2).

**Table A2.** Summary table of the contribution to cc adaptation (developed by the authors).

| | Research | Topic | Contribution to Reduce the Urban Heat Island | |
|---|---|---|---|---|
| Green Roofs | Ciemat and the University of Seville [31] | Green Roofs (Spain) | - | Reduce air temperature by 0.5–2 °C |
| Urban Forest and Parks | A. Dimoudi M. Nicolopoulou [32] | Vegetation in urban environment | - | every 100 m$^2$ of new vegetation in a park reduce 1 °C air temperature |
| | C. Rosenzweig, W. Solecki, et al. [33] | Central Park New York | - | Central Park reduces 2–5 °C the air temperature surrounding Central Park. |
| | Honjo, T., Sugawara H., et al. [34] | Shinjuku Gyoen, Park Tokyo | - | Shinjuku Gyoen Park reduces air temperature by 2 °C |
| | Barbierato, E.; Bernetti I., et al. [35] | Urban Forest in Melbourne | - | Urban Forest reduces urban surface temperatures by 7 °C |
| | The City of Melbourne [36] | Urban Forest in Melbourne | - | City's thermal cameras show Urban Forest reduces urban surface temperatures by 20 °C |
| | **Research** | **Topic** | **Contribution to Runoff Reduction** | |
| Green Roofs | Banting et al., Ryerson University for The City of Toronto [21] | Green Roofs in Toronto | - | Retain 70–90% of runoff in summer and 40–50% in winter |
| Green Façades | Ostendorf M et al., Southern Illinois University Edwardsville, US [37] | Green Wall runoff retention | - | retain 50–75% of runoff, being able to reach 100% |
| Urban GI | Rain City Strategy, the City of Vancouver [38] | Green Urban Infrasructure Strategy | - | Implementing the Rainwater Bulletin the objective is to reduce by 90% the runoff of the city |

*Appendix A.3. Contribution to Human Comfort and Psychology*

This section extracts findings from a series of research that analyze the influence of greenery on human comfort; in terms of acoustic, visual, and climatic comfort. It also includes the psychological benefits derived from the relationship of human being with nature and greenery (Table A3). Nevertheless, the findings show limitations. For instance, an important mass of greenery is needed to achieve significant acoustic attenuation results [39]; the acoustic influence of green roofs is different in wooden or metal frame buildings [40]; transparent green façades may generate glare of discomfort in the interior [41]; and, while greenery contributes to climate comfort by reducing outdoor temperature, it also increases the humidity.

**Table A3.** Summary table of the contribution to human comfort and psychology (developed by the authors).

| | Research | Topic | Contribution to Human Comfort | |
|---|---|---|---|---|
| Acoustic Comfort | Greg Watts et al. [39] | Influence of trees to reduce noise | - | 30 m thick ring of trees reduce traffic noise by 6 dB (A) |
| | Connelly, M.; Hodgson, M. [40] | Influence of green roofs and acoustics | -<br><br>-<br><br>- | 15 cm of green roof in a wooden module reduces<br>5–13 dB (in 2.000 Hz frequencies); and 8 dB (above 2.000 Hz frequencies)<br>in a metallic module it reduces by 10 dB (in low frequencies) and 20 dB (in medium high frequencies) |
| Visual Comfort | Jon Laurenz [41] | Transparent Green Façade and visual comfort | - | Visual discomfort, luminance contrast exceeds 1 to 3 between the observed object and its background |
| Climate Comfort | Summary of Tables A1 and A2 | Contribution of greenery to CC Adaptation and Mitigation | -<br>- | Reduces air temperature 0.5–5 °C<br>Increases humidity by 7–10% |
| | **Research** | **Topic** | **Contribution to Psychology** | |
| Urban Greenery | Stephen Kellert, Edward O. Wilson [42] | Biophlia | -<br><br>- | There is an innate need of the human being to be and feel close to nature<br>Observing nature produces a relaxation and mental liberation that also encourages creativity |
| | Rachel Kaplan & Stephen Kaplan [43] | Experience of Nature | -<br><br>- | people are generally more satisfied in green spaces<br>the presence of nature contributes to a faster recovery from illness |
| | Patrick Mooney [44] | Connecting People and Place | - | interaction with the landscape improves public wellbeing |

Regarding the psychological benefits, research studies highlight the innate need of human beings to be and feel close to nature, in the context of biophilia; as well as how nature produces a relaxation and mental liberation that also encourages creativity [42]. Of note, however, is the generation of feelings of fear, aversion, rejection, and antipathy that nature produces in some people, which should not be overlooked [42]. Overall, people are generally more satisfied in green spaces. In addition, the presence of nature contributes to a faster recovery from an illness [43], and interaction with the landscape improves public wellbeing [44].

*Appendix A.4. Contribution to Air Quality*

The contribution of GUIS to air quality is mainly based on the ability of vegetation to purify and trap polluting particles from the environment.

The paper analyzes a series of research studies (Table A4) to extract the following data: urban forest purifies between 0.3and 1% of air pollutants [23], green roofs capture 0.35 mg of CO, 1.6 mg $NO_2$, 3.14 mg of $O_3$, 2.17 mg of PM10, and 0.61 mg of $SO_2$ (for every 101 ha of green roof) [21]. Furthermore, inside buildings, vegetation helps to reduce the concentration of volatile pollutants such as benzene, formaldehyde, or toluene [45].

**Table A4.** Summary table of air quality (developed by the authors).

| | Research | Topic | | Contribution to Air Quality |
|---|---|---|---|---|
| Urban Forest | McPherson, E.G. + Nowak, D.J. et al. [23] | Chicago's Urban Forest | - | Reduces air pollutants of Chicago by 0.3–1% |
| Green Roofs | Banting et al., Ryerson University for The City of Toronto [21] | Green Roofs in Toronto | - | Green roofs trap, 0.35 mg of CO; 1.6 mg $NO_2$; 3.14 mg of $O_3$; 2.17 mg of PM10; and 0.61 mg of $SO_2$ |
| Green Walls | Soreanu, G.; Dixon, M.; Darlington, Alan [45] | Indoor Green Walls | - | Reduce the concentration of volatile contaminants such as benzene, formaldehyde or toluene |

*Appendix A.5. Contribution to Urban Biodiversity*

In the urban context, Schaefer argues that urban biodiversity faces significant challenges of loss of urban habitat and water, air, and soil qualities [24]. Previous authors had already pointed out the need to integrate nature in designs and in our lives in the city, such as Ian McHarg (1969) [46] and Michael Hough (1995) [47]. More recently, the approach of Ron Kellett and Cynthia Girling [48] remarks that there is the need not only of preserving the ecological structures of the city, but also the need of connecting green spaces through green corridors, to promote urban biodiversity. GUIS would contribute significantly in connecting these green spaces and creating biodiversity corridors within cities (Table A5).

**Table A5.** Summary table of the contribution to urban biodiversity (developed by the authors).

| | Publication | Topic | | Contribution to Urban Biodiversity |
|---|---|---|---|---|
| Urban Biodiversity | Valentin Scahefer et al. [24] | Urban Biodiversity | - | Urban biodiversity faces significant challenges of loss of urban habitat and water, air and soil qualities |
| | Kellett, R.; Girling C. [48] | Skinny Streets and Green Neighborhoods | - | The need to preserve the ecological structures of the city, but also the need to connect green spaces through green corridors) |

*Appendix A.6. Contribution to Urban Agriculture*

The research analyzed reveals how a hydroponic recirculating greenhouse produces between 50 and 100 kg of vegetables per square meter [49]. Considering an average consumption of fresh vegetables of about 100 kg/year, research estimates that, with 5000 hectares of sunny roofs in New York, it would be able to produce the necessary vegetables for 30 million people. In the case of Barcelona, with a population of 1.6 million and an area of 100 $km^2$, it could produce the vegetables necessary for its population using just 2–3% of that area.

Similar research assures that hydroponic or aquaponic plantations can produce between 250–500 tons per hectare per year [50]. Of interest is also how integrated greenhouses in buildings can be used to reuse over production of heating to feed the building's air conditioning/heating system, as well as to reuse the building's excess $CO_2$ to feed the greenhouse plants [51] (Table A6).

**Table A6.** Summary table of the contribution to urban agriculture (developed by the authors).

| | Publication | Topic | Contribution to Urban Agriculture |
|---|---|---|---|
| Urban Agriculture | Caplow, T [49] | Building integrated agriculture | - Hydroponic recirculating greenhouse produces between 50–100 kg of vegetables per square meter. |
| | Proksch, G. [50] | Creating urban agricultural systems | - Hydroponic or aquaponic plantations, can produce between 250–500 tons per hectare per year |
| | Sanyé-Mengual, E. et al. [51] | Rooftop Greenhouse lab | - Greenhouse overheating to feed the building's conditioning system, as well as to reuse the building's excess $CO_2$ to feed the greenhouse plants |

*Appendix A.7. Contribution to Architectural Aesthetics*

This section focuses on emphasizing the expressive capacity that green envelopes represent (green roofs and façades), for the many designers and architects who have already integrated them into their designs. The reflections of internationally renowned architects like Jean Nouvel, who emphasizes that greenery is a material whose shape seems to disintegrate and vanish [52], are remarkable. He also highlights the opportunity that vegetation offers to introduce the dimension of time—or the ephemeral dimension, in Nouvel's terms—to the building [53]. On the other hand, Herzog and de Meuron are attracted to the "scruffy atmosphere" component that the greenery represents in their building on Rue des Suisses in Paris [54].

**Appendix B. Classification of Green Urban Infrastructure Solutions**

This section analyzes different terms used around the world which refer to GUIS and includes a classification of GUIS into different categories.

For instance, in Europe, it is quite common to use the concept of "Nature-Based Solutions" (NBS) and "Re-naturing cities" [55]; as well as Sustainable Urban Drainage (SUD) [56]; or "Urban re-vegetation" [57]. In North America, and more specifically in the West Coast, it is becoming increasingly common to refer to similar concepts using the term "Low Impact Developments" (LID) [58]. Lastly, in Australia, the term of "Water Sensitive Urban Design" (WSUD) [59] is being frequently used.

After analyzing these terms, this paper proposes the term "Green Urban Infrastructure" [60] as the most comprehensive one. It considers that GUIS is the term that most broadly encompasses of all other remaining terms (NBS; SUDs; LIDs; WSUD).

The research classifies GUIS in two major categories: those related to the building scale (Green Building Envelopes); and those associated to a larger urban scale.

*Appendix B.1. Green Urban Infrastructure Solutions at the Building Scale*

Green Urban Infrastructures associated to building envelopes are split into two main construction systems: Green roofs and Green façades.

This classification also includes energy roofs and façades for generating green energy; as well as greenhouse roofs and façades.

Appendix B.1.1. Green Roofs

Green roofs are generally classified into two major subcategories: extensive and intensive [61]. The thickness of the growing medium determines the type of roof: extensive green roofs are considered when the growing medium is moderate, 8–15 cm deep. Green roofs above this range are described as intensive [62].

This classification also includes energy roofs (photovoltaic and wind); and rooftop greenhouse (Figure A1).

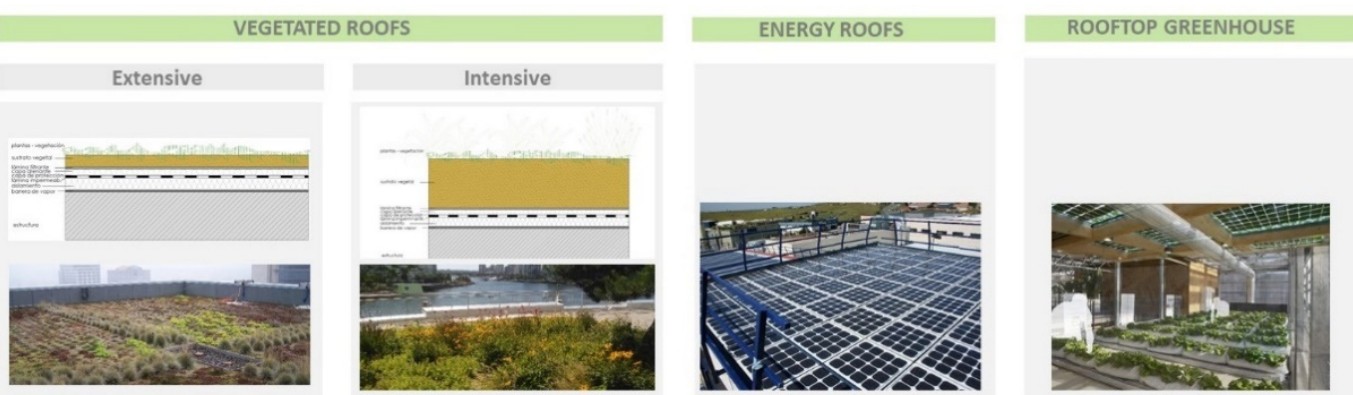

**Figure A1.** Types of green roofs (table and images by the authors).

Appendix B.1.2. Green Façades

Green façades are also divided into two major subcategories: those attached to an opaque wall—*opaque green façades*; and those associated to a transparent wall—*transparent green façades*. These, in turn, are broken down into five subgroups: greenery climbing through an opaque wall; greenery attached to an opaque wall as a green tapestry; greenery composed of green vertical panels; greenery between two transparent layers; and greenery in the external layer and a transparent wall as the internal layer [26].

This classification also includes *energy façades* (photovoltaic and wind) within green façades categories (Figure A2).

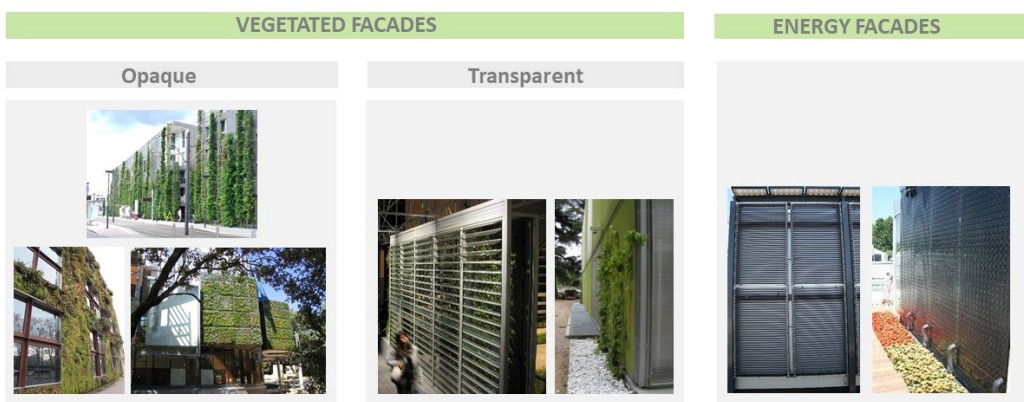

**Figure A2.** Types of green façades (table and images by the authors).

*Appendix B.2. Green Urban Infrastructure Solutions at the Urban Scale*

This section focuses on green infrastructure at the urban scale, leaving out other types of green infrastructure on a territorial or peri-urban scale such as forests, regional parks, agricultural areas, or even urban parks. This classification focuses on the urban scale and proposes to group them into the following four main subcategories: *green canopies; green permeable pavement; green infiltration systems; and bio-retention areas*.

Appendix B.2.1. Green Canopies

This paper distinguishes two major green canopies: *vegetated canopies*, such as urban forest or vegetated pergolas; and *energy canopies*, which generate green energy, such as

photovoltaic canopies or urban wind turbines. The following figure (Figure A3) shows some examples of these types of green canopies.

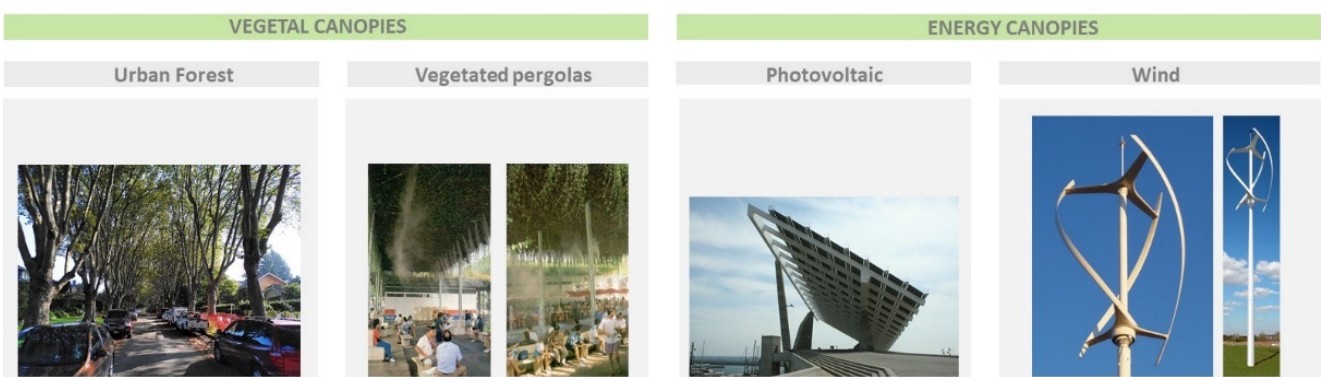

**Figure A3.** Types of green canopies (table and images by the authors, except images of vegetated pergolas [31] and wind turbines [63]).

Appendix B.2.2. Green-Permeable Pavement

This paper considers green pavements both those that allow rainwater to infiltrate through the pavement's material and those in which rainwater infiltrates through the joints [64]. The research, hence, classifies them into two main groups: *Permeable pavements due to their composition*, which can in turn, be modular or continuous; and *Pavements that are permeable at their joints*, which can also be modular or continuous (Figure A4).

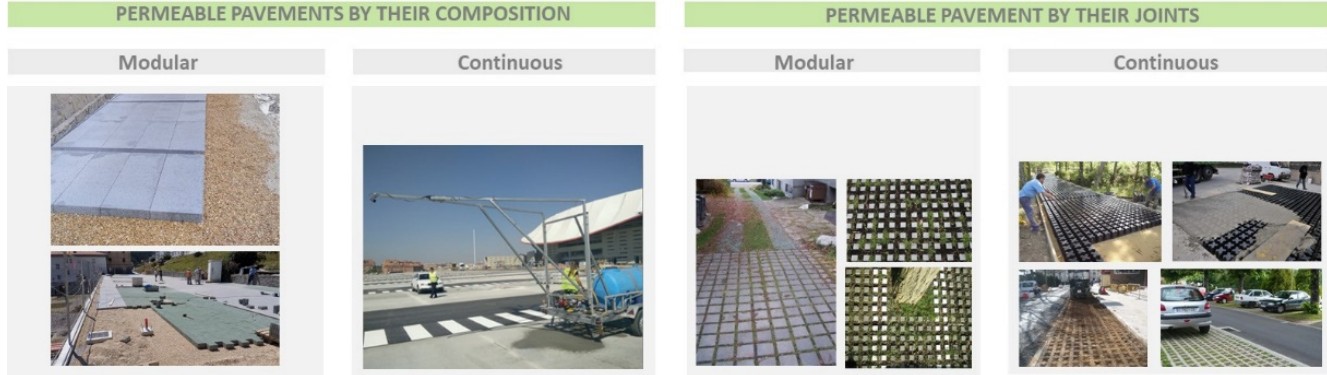

**Figure A4.** Types of green pavement (table and images by the authors, except images of continuous pavements [65,66]).

Appendix B.2.3. Green Infiltration Systems

These systems collect rainwater directly, as well as from adjacent impervious surfaces. They store, purify and subsequently either water infiltrates the ground to recharge aquifers, or it is guided to another infiltration system. Systems are divided into three main groups; *Infiltration trenches basins and swales; Filtration trenches and drains;* and *Accumulation and infiltration tanks* [67] (Figure A5).

## SISTEMAS DE INFILTRACIÓN VERDE / GREEN INFILTRATION SYSTEMS

| INFILTRATION TRENCHES, BASINS AND SWALES | FILTRATION TRENCHES AND DRAINS | ACCUMULATION AND INFILTRATION TANKS |
|---|---|---|

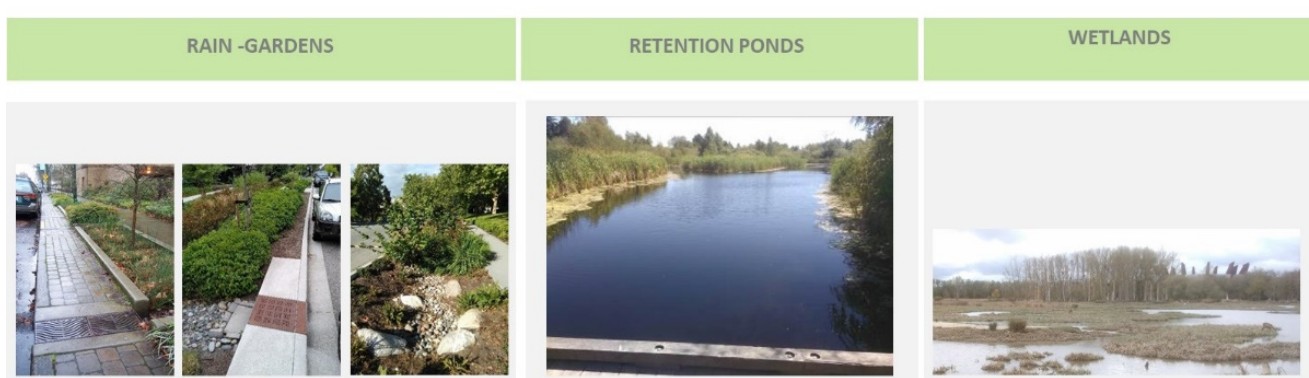

**Figure A5.** Types of green infiltration systems (table and images by the authors, except the first image of infiltration trench [67]).

Appendix B.2.4. Bio-Retention Areas

Bio-retention areas are depressions in the ground that retain rainwater and runoff from adjacent impervious surfaces and are generally covered with vegetation, trees or even by the retained water itself. Subsequently, the water can directly infiltrate the ground or be guided to another infiltration system. There are three main types of bio-retention areas: *rain gardens; retention ponds;* and *wetlands* [67] (Figure A6).

## ÁREAS DE BIO-RETENCIÓN / BIO-RETENTION AREAS

| RAIN -GARDENS | RETENTION PONDS | WETLANDS |
|---|---|---|

**Figure A6.** Types of bio-retention areas (table and images by the authors).

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
