# Peer review of "Green Urban (RE) Generation: A Research and Practice Methodology to Better Implement Green Urban Infrastructure Solutions"

_land, doi:10.3390/land10121376_

Round 1

Reviewer 1 Report

This paper suggests that green infrastructure implementation must be scaled to meet SDGs, but there has been resistance to implementation of green infrastructure. The authors propose a method of community engagement, CDM, as a tool to drive implementation. They discuss a community engagement process implemented with the Life Good Local Adapt European Project. They also discuss typologies of green infrastructure, benefits of green infrastructure, and examples of projects. The body of the paper does not focus on the CDM, even though the initial presentation and conclusion seems to indicate that this is the main subject of the research. The primary research question of the paper and the methods used to answer the question are not clear. The conclusions are not clearly linked to the findings.

This paper is motivated by a global need to drive uptake and implementation of green infrastructure. But, the method is very local. How does this local method connect to the larger problem? Does the local method apply in all places that would benefit from using green infrastructure? If they all used this method, would that drive sufficient quantities of green infrastructure to meet goals? It would be helpful to include more discussion about what using the CDM can and cannot accomplish.

Section 3, the classification of green infrastructure solutions, should be removed. It veers off the main topic, which I believe should be the CDM. Green infrastructure types are also likely be well known by prospective readers. If the authors feel it is necessary to retain, perhaps consider moving the information into an appendix.

Section 4 should be the primary focus of the paper. This can be explained in much more depth. Was the engagement process different in at all among the three cities? Are the participants representative of the demographics of the city? It would be helpful to focus more on exactly what happened, especially given the paper’s assertation (or somewhat implicit assertion) that this is a demonstration of a replicable framework.

How do the stated engagement goals (lines 283-290) relate to the larger identified goals of improving uptake of green infrastructure? How will you know if you’ve met these goals? Clarifying the goals of the CDM across the whole paper and methods for evaluating effectiveness would greatly strengthen the paper.

The text from 313-464 should not be bulleted. This is the most important work in the paper and there are a lot of interesting details that could be uncovered and discussed further. There needs to be improved description of most aspects of the project. For example, a screenshot of Citizen and commentary on its use may be valuable. The main findings of the engagement process briefly described in the paragraph beginning on line 414 requires far more attention with greater detail of the findings and the shortcomings.

Section 5 is material that would be standard to include in an introduction or perhaps a background section motivating the paper. The paper would flow much better if this were highly condensed and moved into the introduction. Tables 5-9 should be eliminated.

Section 6 should be directly linked to Section 4.

All of the community engagement work (Sections 4 and 6) should be explicitly aligned with the suggested framework in CDM: research, community engagement, implementation, repeat. Right now, it is difficult to understand how this is a demonstration of the CDM framework. Reorganizing the structure of the presented community work would help develop the link between this work and the conclusions.

Reviewer 2 Report

attached document

Reviewer 3 Report

This paper will be a great resource to urban managers who are considering implementing green infrastructure. The tables are especially helpful. One thing that is not discussed is the barrier of financing GUIs especially if there are no national or state mandates and programs to offer funding. Start-up costs are always very high with GUIs. If the authors can convert some of their bullet lists to narrative form, that would be best.

Round 2

Reviewer 1 Report

This draft is significantly improved over the previous draft. It is much more focused and much clearer. It was successful to move material to appendices. To be of publishable quality, however, the study requires several amendments.

The primary issue is that it is not clear the CDM leads to—as the title suggests— “better” implementation. How do the authors know for certain this is true? What is the standard route of implementation? What are the measures for “better”? In what ways has the CDM improved performance? How should performance be assessed?

Currently, the study seems to tell a story of what happened in three related projects, but it is difficult to extract larger lessons. How does the CDM framework build from existing literature on implementing GUI? How can these lessons be applied to new places? What would need to change from current practice to implement the CDM and what would be the expected impact of doing so?

Finally, I started to make grammatical comments, but then stopped. The paper requires further copy editing.

Some line-item comments follow:

11-13, 20

Remove figure reference and citations from abstract. Revise text/typos.

31-32

Maybe an example or two from a developing city, rather than three reasonably similar cities? (Vancouver is more emblematic of parts of North America than the Americas)

44

There is an extra comma

44

More citations would be useful here. My understanding is there is some debate over whether GUI has been quite as cost effective as hoped. It seems over-simplifying to make this statement, but perhaps there is wider agreement than I realize.

50

Delete the semi-colon

51

International is misspelled

55

Can you be more explicit about the link to climate change? I assume it matters what type of GUIs and that not all address climate change equally. This statement would be stronger and more interesting if it were expanded and integrated with some of the references that are now in Appendix A.

56

barriers

Fig 2

These phases don’t quite make sense to me. Either: show 3 phases, not the 3 phases two times, or rename the second set of phases. Also, anyone implementing the CDM need to do the first phase of research? Or have you already done that now? Or perhaps it is necessary to identify the GUI relevant to the site?

124-125

This either merits more explanation. Why does citizen engagement lead to more effective GUI?

331

Switch fractions to percentages to be consistent with other figures

538

This does not read well as a bulleted list. In addition, the first sentences don’t quite make sense. For example, “Vegetated canopy” is not a lesson learned and this discussion should be expanded. All of section 5.3. should be reorganized and edited for clarity. The lessons learned are critical.

594

Integrate this sentence into a paragraph
